# Accelerated absorption of regular insulin administered via a vascularizing permeable microchamber implanted subcutaneously in diabetic *Rattus norvegicus*

Leah V. Steyn[1], Delaney Drew[1], Demetri Vlachos[1], Barry Huey[1], Katie Cocchi[1], Nicholas D. Price[1], Robert Johnson[2], Charles W. Putnam[1], Klearchos K. Papas[1,2]*

1 Institute for Cellular Transplantation, Department of Surgery, University of Arizona College of Medicine-Tucson, University of Arizona, Tucson, AZ, United States of America, 2 Procyon Technologies, LLC., Medical Research Building (Room 121), University of Arizona, Tucson, AZ, United States of America

* kkpapas@surgery.arizona.edu

**Data Availability Statement:** All relevant data are within the manuscript.

**Funding:** K.K.P. 2-SRA-2020-870-S-B Juvenile Diabetes Research Foundation www.jdrf.org The

## Abstract

In Type 1 diabetes patients, even ultra-rapid acting insulins injected subcutaneously reach peak concentrations in 45 minutes or longer. The lag time between dosing and peak concentration, as well as intra- and inter-subject variability, render prandial glucose control and dose consistency difficult. We postulated that insulin absorption from subcutaneously implantable vascularizing microchambers would be significantly faster than conventional subcutaneous injection. Male athymic nude *R. norvegicus* rendered diabetic with streptozotocin were implanted with vascularizing microchambers (single chamber; 1.5 cm$^2$ surface area per side; nominal volume, 22.5 µl). Plasma insulin was assayed after a single dose (1.5 U/kg) of diluted insulin human (Humulin®R U-100), injected subcutaneously or via microchamber. Microchambers were also implanted in additional animals and retrieved at intervals for histologic assessment of vascularity. Following conventional subcutaneous injection, the mean peak insulin concentration was 22.7 (SD 14.2) minutes. By contrast, when identical doses of insulin were injected via subcutaneous microchamber 28 days after implantation, the mean peak insulin time was shortened to 7.50 (SD 4.52) minutes. Peak insulin concentrations were similar by either route; however, inter-subject variability was reduced when insulin was administered via microchamber. Histologic examination of tissue surrounding microchambers showed mature vascularization on days 21 and 40 post-implantation. Implantable vascularizing microchambers of similar design may prove clinically useful for insulin dosing, either intermittently by needle, or continuously by pump including in "closed loop" systems, such as the artificial pancreas.

## Introduction

In response to perturbations of plasma glucose concentration and other physiologic cues, the normal endocrine pancreas modulates its release of insulin (and other hormones); insulin,

funders had no role in study design, data collection and analysis, decision to publish, or preparation of the manuscript.

**Competing interests:** I have read the journal's policy and the authors of this manuscript have the following competing interests: [R.J. and KKP are co-founders of and have financial interests in Procyon Technologies LLC.]. This does not alter our adherence to PLOS ONE policies on sharing data and materials.

entering the portal venous circulation in approximately five second pulses and longer oscillations, rapidly and precisely maintains glucose homeostasis [1]. In Type 1 diabetes mellitus (T1DM), the endocrine pancreas fails to produce insulin in sufficient quantities, if at all [2]. Although pharmacologic insulins are a life-saving therapy, they fail to mimic the rapidity and precision of glucose modulation conferred by the normal pancreas [3]. Consequently, despite insulin therapy, diabetic patients remain vulnerable to a range of costly, long-term, disabling complications.

Pharmacologic insulins are most often injected into the subcutaneous (SC) tissue of the abdominal wall or extremities; however, because of the relative avascularity of SC tissue [4], the dispensed insulin—depending upon its formulation—may require an hour or longer to attain its peak concentration in the blood [4]. This delay in absorption renders insulin therapy an exercise in predictive dosing, especially when anticipating the substantial swings in blood glucose concentration encountered during and following meals [5]. The lag time between SC dosing and achieving an effective insulin concentration in the blood also blunts the precision otherwise afforded by continuous glucose monitoring (CGM) devices, sophisticated dosing decision algorithms, precision insulin pumps, open- or closed-loop systems, and the "artificial pancreas" [4, 6].

Efforts to accelerate absorption of insulin have pursued two general approaches: novel formulations of pharmacologic insulins and altered routes of administration. One or more amino acid changes of the insulin molecule itself [7–12] and/or the addition of chemicals [13] reduce the tendency of native insulin to form multimeric complexes which first must dissociate into monomers or dimers in order to be absorbed [14]. These modifications of pharmacologic insulin have yielded only incremental increases in the rate of absorption (reviewed in [4, 15]). Consequently, ultra-rapid insulins have not achieved wide-spread patient acceptance [4], in part because of increased drug costs [15] and a greater frequency of injection site reactions [16].

The second approach–alternative routes of administration affording more rapid uptake of insulin—have included intradermal microinjection [17] or "jet spray" [18, 19], intraperitoneal instillation [20, 21], or the inhalation of specially formulated insulins [22, 23]. None of these strategies has gained widespread clinical acceptance, in part because of unpredictable dosing, increased complication rates, and patient hesitancy [24, 25].

Although the lag times between SC injection and peak time ($T_{max}$) have been somewhat foreshortened by the development of "ultra-rapid acting insulins" [Faster aspart (Fiasp), 63 min, URLj (Lyumjev), 57 min [15]], the subcutaneous route is beleaguered by a second confounding problem: intra-patient variability of absorption of the specified dose [26–28] resulting in unanticipated blood glucose effects. This variance between anticipated and actual effect upon blood glucose concentration contributes to the incidence of both diabetic ketoacidosis (DKA) and importantly, hypoglycemic episodes [6], especially in those individuals who are constitutively "hypoglycemia unaware" [29, 30].

Unpredictable discrepancies between the intended and evinced biochemical effects of a particular SC dose are multifactorial. The vascularity of the subcutaneous space is not homogenous even within individuals [22, 31–35] and the SC composition is further altered by repeated injections [36, 37]. In a 2018 review [38], the multitude of factors affecting absorption of SC insulin are discussed in detail; they include, among many others: variations in the depth of needle penetration, including inadvertent intramuscular injection; dislodged catheters or mechanical issues associated with insulin pump infusion sets [39]; and seepage of drug from the injection site. It stands to reason that intra-patient variability compounds inter-patient dosage algorithms; predictions and comparisons are therefore fraught, requiring complex modeling [40, 41].

To address the challenges posed by lagging absorption of even ultra-fast insulins and the inconsistency of intrasubject dosing, we evaluated in a diabetic *R. norvegicus* model, subcutaneously implantable vascularizing microchambers, IVMs (Fig 1A). These microchambers–of various footprints and capacities–are engineered of polytetrafluoroethylene membranes (PTFE) and have a surface configuration with large pores ($\sim$ 5 μm) which promotes angiogenic ingrowth; the inner, membrane, which has much smaller pores, serves to block host cells from entering and occupying the microchamber [42, 43]. We report here that the IVM, implanted in the SC space, accelerates the absorption of inexpensive regular human insulin, achieving pharmacokinetic profiles potentially superior to ultra-rapid acting insulins. Importantly, our data also suggest that the IVM improves consistency of insulin delivery, thereby potentially avoiding hazardous deviations in blood glucose concentration.

## Materials and methods

### Ethical care of experimental animals

All experiments were performed with the written approval of and in accordance with the guidelines established by the Institutional Animal Care and Use Committee at The University of Arizona.

Anesthesia for procedures was provided with inhaled isoflurane with oxygen; SR Buprenorphine, which provides approximately 72 hours of pain relief, was included for analgesia. To alleviate suffering, analgesics (SR Buprenorphine) were provided for 72 hours postoperatively. The animals were observed daily and weighed weekly; any animal showing signs of severe distress or a body weight loss exceeding 20% was removed from the study and immediately euthanized. The Body Condition Score (BCS) and the Early Removal Score (ERS) were utilized to assess the overall health of the rodents and to communicate this information to University

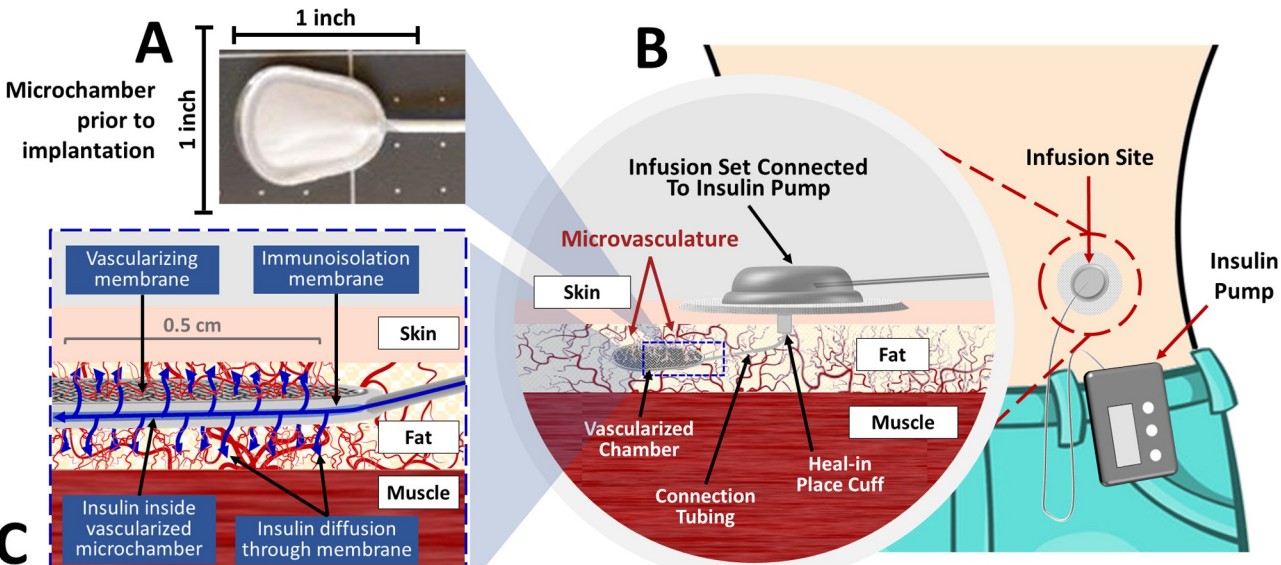

**Fig 1. The Procyon implantable vascularizing microchamber (IVM) and an illustrative clinical application.** (A) The vascularizing, microchamber (depicted before implantation), fabricated of PTFE, has a surface area per side of 1.5 cm$^2$ and a nominal volume of 22.5μL. (B) One potential clinical application of the IVM is to couple it with an insulin pump. The *infusion set* connects the *insulin pump* to the SC implanted *microchamber*; the IVM's *connection tubing* is fitted with a *heal in place cuff* fabricated from material that promotes vascularization. (C) A schematic of the vascularized microchamber, depicting the ingrowth of blood vessels to its surface and the flow of insulin from within the microchamber, through the permeable membranes, and into the richly vascularized SC space surrounding the IVM.

Animal Care (UAC) veterinary staff. Rats were euthanized using American Veterinary Medical Association (AVMA) approved methods, notably overdose of inhaled CO2 or isoflurane in oxygen. Secondary methods to assure death were also used; these included exsanguination, bilateral thoracotomy, or vital organs harvest.

## Animal subjects

Male athymic nude *R. norvegicus* between 9–12 weeks of age and weighing 250–350 g were purchased from Envigo (Livermore, CA, USA). Because estrogen interferes with streptozotocin [44] (see next), only male animals were included. Athymic nude rats were chosen to abrogate a possible antibody response against the human insulin to be injected [45]. Animals were housed for a seven-day acclimatization period before initiating any experimental procedures.

## Induction of diabetes

To avoid endogenous insulin in subject rats from cross-reacting in the human insulin assays, animals were rendered diabetic by a single intraperitoneal injection, 60 mg/kg, of streptozotocin (Sigma-Aldrich, St. Louis, MO, USA) [44]. Blood glucose concentrations were monitored daily using a Freestyle Lite Glucose Monitoring System (Abbott Diabetes Care Inc., Alameda, CA, USA). Rats were considered diabetic when blood glucose concentrations greater than 200 mg/dl were recorded on three consecutive days. Rats verified as diabetic were maintained with exogenous long-acting insulin (Lantus, Sanofi-Aventis, Bridgewater, NJ, USA) throughout the study; maintenance insulin treatment also serves to forestall insulin resistance [46]. Maintenance insulin therapy was suspended for 24 hours prior to each insulin kinetics study.

## Implantation of vascularizing microchambers (IVMs)

Two days after the initial insulin-kinetics assay two empty, 1.5 cm$^2$ per side, single-chamber (having a nominal internal volume of 22.5 μl), implantable vascularizing microchambers, IVMs (Procyon Technologies LLC, Tucson, AZ, USA) (Fig 1A) were implanted subcutaneously on the dorsal aspect of each rat.

## Insulin kinetics assays

**Subcutaneous injection of human insulin.** The initial insulin kinetics assays were performed 7 days after STZ induction of diabetes. Nonfasted rats received a single subcutaneous injection of insulin human, 1.5 U/kg (Humulin R; Lilly, Indianapolis, IN, USA). Each dose of insulin was diluted in 0.5 mL of sterile saline (0.9%) solution, then injected subcutaneously using a glass Hamilton syringe fitted with a 20-gauge sharp needle.

**Blood sample collection.** Blood samples (150 μl) were collected from the tail vein before injection (time = 0), and at 5-, 15-, 30-, 60-, 90-, and 120-minutes following injection. Blood samples for insulin assays were stored on ice in 1.5 ml centrifuge tubes with EDTA until separation of plasma by centrifugation at 13,300 x g for 15 minutes at 4˚ C. Plasma samples were stored at -20˚ C.

**Administration of human insulin via subcutaneously implanted IVMs.** On day 28 post-implantation, each rat received a single dose of insulin human, Humulin R, 1.5 U/kg (Lilly, Indianapolis, IN, USA), diluted as described above. The insulin was then instilled into one of the two IVMs via its port. Blood samples for plasma insulin were collected and processed as described above.

**Analysis of rat plasma samples for human insulin concentration.** Plasma samples collected from each insulin kinetics assay were analyzed for human insulin concentration using a

commercial human insulin ELISA kit (Alpco, Salem, NH, USA). Samples were run in duplicate. The ELISA kit has a detection range of 3.0–200 μIU/ml and a sensitivity of 0.399 μIU/ml. The measured insulin concentrations were converted from IU/ml to μg/ml according to the manufacturer's recommended conversion ratios of 1 IU of human insulin = 6nmol = 34.8 μg of insulin.

### IVM retrieval and histopathology

At the conclusion of the study (day 40), rats were euthanized, and the IVMs were retrieved for histology. Additional specimens were obtained 7 and 21 days after implantation from rats in a cohort not included in the insulin analyses reported herein. After removal, each IVM including its adherent tissue was placed in 10% neutral buffered formalin for fixation, then processed for paraffin embedding. Paraffin blocks were sectioned and 5 μm slices were collected throughout the microchamber. Sections were stained with hematoxylin and eosin and imaged with a Keyence BZ-X710 microscope (Keyence Co, Osaka, Japan).

### Statistical analysis

**Insulin kinetic curves.** The concentrations of human insulin in each rat were normalized to zero by simple subtraction of the variance at t = 0 from all values for that animal. The insulin kinetic curves were created by calculating the mean values and standard deviations at each time point. In several instances, complete plasma insulin concentration data were not obtained for a given rat in a particular assay; if more than three of the seven data points were unavailable, the subject was excluded from analysis. Data from one animal in the subcutaneous injection (control) group was excluded from analysis on that basis.

**Peak insulin times and concentrations.** The mean peak times ($T_{max}$) and concentrations of the insulin kinetic assays were obtained as follows: for each subject the greatest plasma insulin concentration was identified, and both the time point (*e.g.*, 10 min) and the insulin concentration at that time point were tabulated. The mean and standard deviation (SD) for each category were then calculated. The two assays, subcutaneous injection at day -2, and IVM administration on day 28, were compared by the unpaired, two-tailed, t-test. The full kinetic curves were compared with the nonparametric Friedman's test [47]. Statistical tests were conducted with GraphPad Prism version 9.4.1 for Windows (GraphPad Software, San Diego, California USA, www.graphpad.com).

## Results

### Insulin-kinetics study after subcutaneous (SC) injection in diabetic rats

Five days after successful induction of STZ diabetes, rats were injected subcutaneously with diluted human insulin (Humulin R), followed by blood sampling for plasma insulin concentration over a two-hour period. Consonant with the extensive literature documenting inter-subject variability of insulin absorption after SC injection, the individual insulin curves were quite variable in timing and concentration between animals (Fig 2A), with no clear peak by visual inspection (Fig 2C). The mean peak of plasma insulin–$T_{max}$–was calculated as 22.7 (n = 11, SD 14.2, 95% CI 13.18–32.27) min post-injection and the peak concentrations averaged 12.3 (SD 3.16) ng/ml (Table 1). In the experiments described above, the Humulin R stock suspension was diluted in 500 ul sterile saline, approximately a 100-fold dilution. Because the rate of insulin absorption directly correlates with the concentration of mono- and dimers, dilution of the stock solution might seem to favor dissociation of multimeric insulin [9, 14]. In another experiment we compared two concentrations–diluted versus stock–of Humulin R

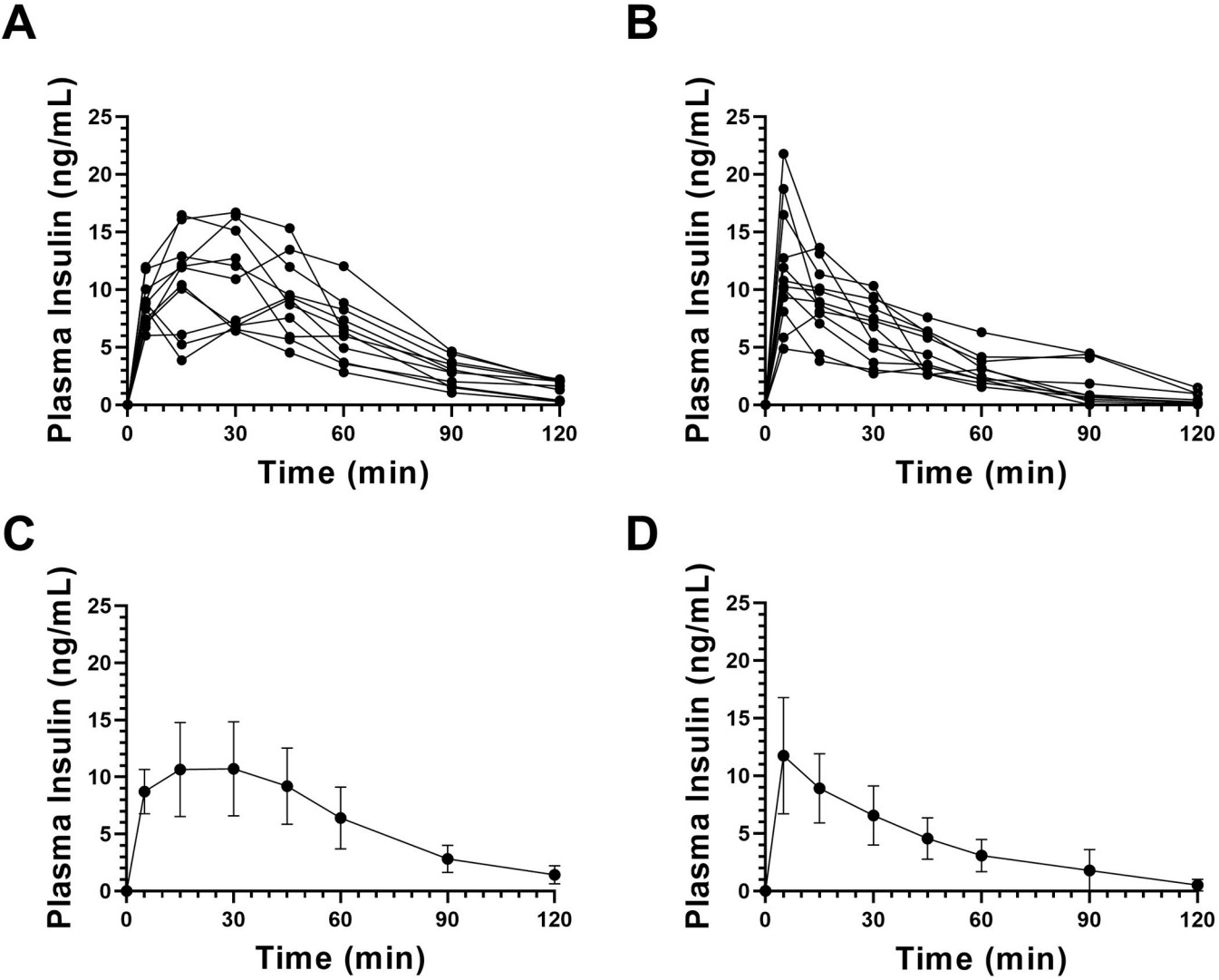

**Fig 2. Plasma insulin kinetics after either SC or IVM injection of regular insulin.** Plasma concentrations of human insulin were collected over 120 minutes following either subcutaneous injection (A,C) or instillation into subcutaneous IVMs 28 days after implantation (B,D) of regular insulin human (Humulin R) in identical doses. Shown in A and B are the data for the individual subjects in each group of rats; in C and D are the insulin kinetic curves generated by plotting means and standard deviations (SD); n = 11 rats in the SC injection group (one rat was excluded from the analysis because of incomplete plasma insulin concentration data); n = 12 rats in the IVM group.

insulin after SC injection. The plasma insulin curves did not vary significantly between the two concentrations (unpublished data), confirming that dilution of Humulin R did not *per se* affect its rate of absorption.

## Insulin kinetics after administration via a subcutaneous microchamber

The implantable vascularizing microchamber (IVM) by Procyon (Fig 1A) is designed to provide a means for rapid, consistent absorption of insulin without repeated needle (or pen or pump cannula) injections; it measures 1.5 cm² in surface area per side. Its surface is engineered to promote rapid vascularization. Published reports suggested that to be effectively vascularized, subcutaneously implanted devices typically require three to four weeks in mice [48, 49], two to four weeks in rats [50], and two months or less in primates [51]. We therefore

**Table 1. Mean peak times and concentrations of plasma insulin after subcutaneous or IVM injection in diabetic rats of insulin human (1.5 U/kg body weight).**

|  | Subcutaneous injection | IVM injection on day 28 |
|---|---|---|
| Number of subjects (n) | 11 | 12 |
| Body weight (g) | 302.1 (SD 25.1) | 364.9 (SD 29.6) |
| $T_{max}$ (min) | 22.7 (SD 14.2) | 7.50 (SD 4.52) ** |
| 95% CI of $T_{max}$ (min) | 13.18–32.27 | 4.626–10.39 |
| Insulin concentration (ng/ml) | 12.3 (SD 3.16) | 11.7 (SD 4.97) NS |

Given are means, standard deviations (SD), and 95% Confidence Intervals (CI). Although the peak concentrations are not significantly different (NS), the interval to peak concentration ($T_{max}$) was significantly shorter (** p = 0.0020, by the unpaired, two-tailed t test) when insulin was delivered via microchamber.

conducted the insulin kinetics assays at 28 days after implantation, anticipating mature vascularization by that time. Diluted insulin human was instilled into the IVM at the same dose by body weight as in the SC injection control studies, described above.

At 28 days after implantation, 10 of the 12 individual plasma insulin curves peaked just five minutes after injection (Fig 2B), as did the mean value (Fig 2D). The $T_{max}$ for insulin (Table 1) was calculated as 7.50 (SD 4.52, 95% CI 4.626–10.37) min versus 22.7 (SD 14.2) min with conventional SC injection, a statistically significant difference (p = 0.0020, Table 1). Not only was uptake of insulin from the microchamber accelerated but the inter-subject variability appeared reduced (Fig 2B); this was also attested to by the standard deviations of the mean peak (Table 1) and of the early time points of the insulin curve (Fig 2D). Together, the data at day 28 after implantation, indicate that insulin absorption from the IVM is significantly accelerated with improved inter-subject consistency, compared to conventional SC needle injection.

## Evidence of vascularization revealed by histopathology of retrieved IVMs

The peak insulin data presented in the preceding section strongly support the notion that vascularization of the SC tissues surrounding and invading the vascularizing membranes of the microchambers had matured by 28 days and had thereby facilitated the very rapid uptake of insulin. If so, histologic evidence of blood vessels immediately adjacent to the microchambers is likewise expected in the same time frame.

To examine this supposition, IVMs implanted subcutaneously in rats for 7, 21 and 40 days were retrieved and examined histologically. Although the 1.5 cm$^2$ microchambers were identical to the ones used for insulin kinetics studies, they had not been injected with insulin to avoid confounding effects on morphology that might be induced by the diffusion of human insulin into the surrounding rat subcutaneous tissues. At 7 days after implantation (Fig 3A) a few vessels can be identified but by day 21, subcutaneous vessels in proximity to the vascularizing membranes are both prominent and plentiful (Fig 3B). Images of subcutaneous tissue surrounding the microchambers at 40 days (Fig 3C-3F) reveal the presence of mature vasculature structures with larger vessels quite close to the inner membranes of the IVMs. There was no histologic evidence of fibrotic overgrowth, even at the latest time point. Thus, the histopathology of explanted IVMs indicates that vascularization begins soon after implantation; the numbers of blood vessels greatly increase by three to about six weeks after implantation, an observation congruent with the increased rapidity of insulin uptake on day 28.

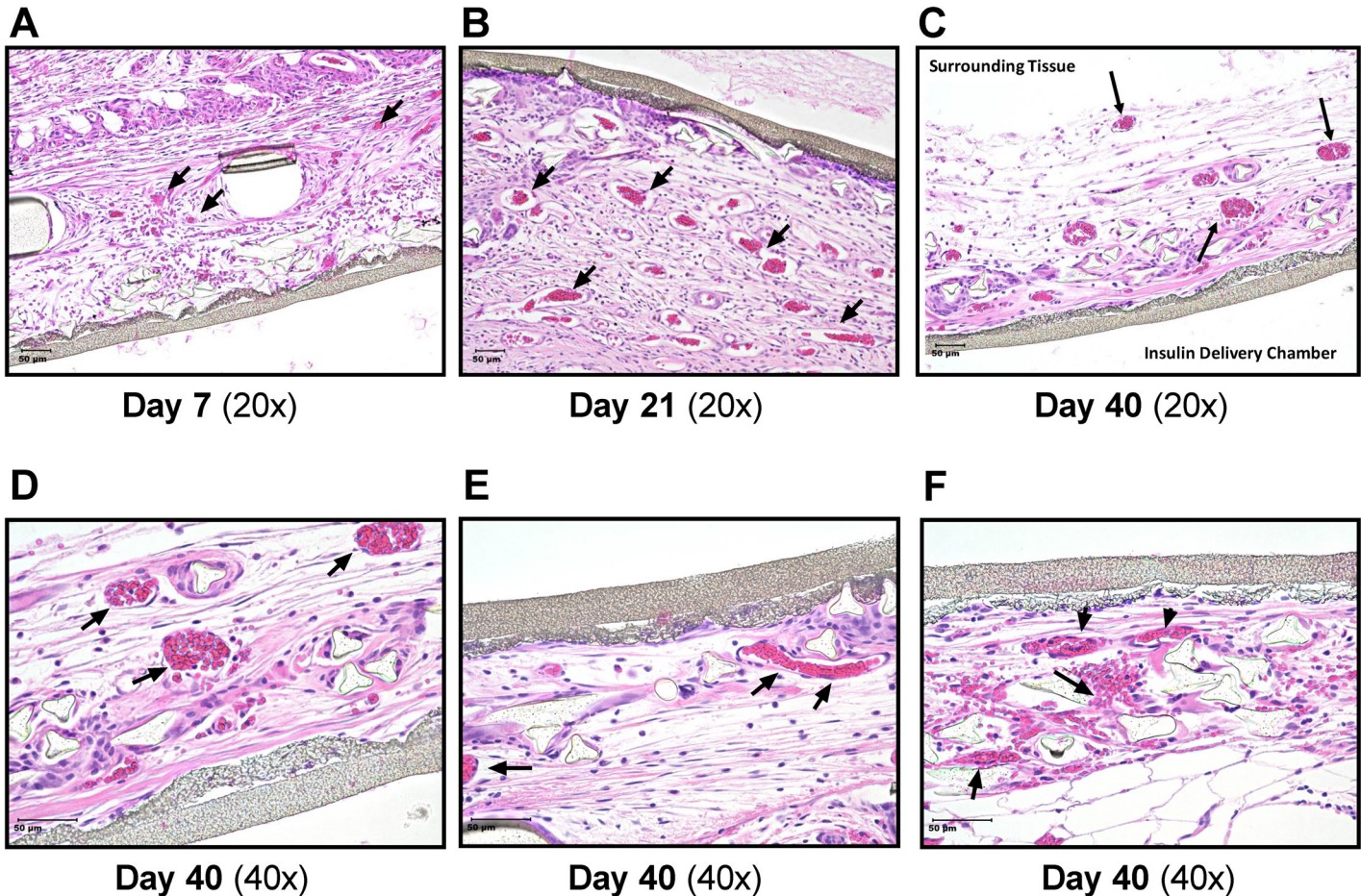

**Fig 3. Histopathologic images of retrieved microchambers and the surrounding subcutaneous tissues.** The images of H & E (hematoxylin and eosin)-stained sections, using 20X (A-C) and 40X (D-F) objectives, of IVMs retrieved 7, 21 or 40 days after implantation. The arrows point to vascular structures. For orientation purposes, the image in C is labeled to identify the *surrounding* (SC) *tissue* and the *insulin delivery chamber*. The linear structure (gray) is the inner membrane of the IVM.

## Discussion

Here we report that a small (1.5 cm$^2$ surface area per side), specially engineered PTFE, vascularizing microchamber (Fig 1) implanted subcutaneously facilitates more rapid uptake of insulin than conventional SC injection. The outer membrane of the IVM has large pores ($\sim 5\mu m$) which allow the ingrowth of blood vessels [42, 43]; this surrounding vasculature fully matures by about one month (Fig 3). Regular insulin human (Humulin R), when instilled into the IVM 28 days after implantation, achieves peak plasma concentrations significantly faster than an identical dose delivered by conventional SC injection, 7.50 versus 22.7 min (Fig 2 and Table 1). Additionally, inter-subject variability was reduced (Fig 2A and 2B). We therefore conclude that the subcutaneously implanted microchambers are functionally mature within weeks, not months.

The subcutaneous space is favored for insulin injection for patient convenience: it is a large space readily accessible for injection and easily examined for problems at the sites of skin penetration. Physiologically, however, it is less than ideal; its vascularity is both sparse and heterogenous. Moreover, repeated injections may generate inflammation and consequent fibrosis, rendering the microenvironment even less conducive to absorption of injected insulin. Finally,

utilizing the SC space for insulin delivery currently requires repeated injections, by needle, pen or pump, their frequency depending upon the chosen insulin formulation(s) and the means of injection.

The primary clinical scenario for which we envisage the IVM achieving clinical utility for T1DM patients (Fig 1) is a conceptual extension of the experiments described in this report–namely, implantation in diabetic patients to facilitate the very rapid uptake of regular insulin. One might argue that accelerating absorption via an IVM is an unnecessary redundancy now that ultra-rapid acting insulins are available. However, the very rapid peak plasma insulin concentrations achieved with regular insulin via IVM more closely mimic the normal human pancreas and are likely superior to even ultra-rapid acting insulins injected SC [16, 52, 53]. The insulin kinetic curves (Fig 2B and 2D) are compatible with an "inject-eat continuum" strategy: this would, first, synchronize the insulin peak with the prandial requirement; and second, avoid the pitfalls of inopportune delays of a meal by simply postponing the injection. As shown in Fig 1, an insulin pump is easily connected to an IVM, ensuring rapid uptake of the pumped insulin; this would allow greater precision in dosing and rapid uptake of prandial boluses. Other challenging clinical scenarios would likewise benefit from rapid uptake of insulin and therefore better glucose control. One example is insulin therapy of diabetics during pregnancy, where precise glucose homeostasis is vital to the well-being, both immediate and long-term, of mother and fetus [54].

Another argument sometimes made is that rapid uptake of insulin confers more risk than benefit, for example, by triggering hypoglycemic episodes. However, the clinical safety and efficacy, especially in post-prandial control, of ultra-rapid acting insulins is now supported by many studies [15, 16, 55, 56]. The factitious "too rapid" assertion also deflects the many advances in continuous glucose monitoring (CGM) devices, insulin dosage-predicting algorithms, precision-dosing insulin pumps, open and closed loop systems, and the "artificial pancreas" (AP) [6]. To this point, a recent, comprehensive review of multiple input AP systems [57], identified two persistent challenges: lag times in acquiring blood glucose concentration data, and delays in absorption of insulin from SC depots. Our study is relevant to the latter concern, which echoes an earlier sentiment: for APs (and similar approaches) to achieve their full potential, the "applied insulin should induce ideally an instantaneous effect" [4]. The IVM potentially offers a clinical means of further closing the gap between insulin dose and blood glucose response.

## Conclusion

Regular insulin human instilled into SC implantable vascularizing microchambers (IVMs) implanted 28 days previously in diabetic rats attains mean peak plasma insulin concentrations in 7.5 (SD 4.50) min, versus 22.7 (SD 14.2) min following conventional SC injection. Intersubject variability was likewise reduced. Histologically, mature vascularization was evident at 21 and 40 days after implantation, indicating that neovascularization surrounding the IVMs was a significant contributor to the rapidity of insulin uptake. We suggest that the implantable vascularizing microchambers reported herein have clinical potential for the painless, repetitive, reproduceable delivery of regular insulin with an absorption rate potentially exceeding that of ultra-rapid acting insulins.

## Acknowledgments

The authors wish to acknowledge Chan A. Ion and Jennifer Kitzmann-Miner for their assistance in preparing graphic work.

## Author Contributions

**Conceptualization:** Robert Johnson, Klearchos K. Papas.

**Data curation:** Leah V. Steyn.

**Formal analysis:** Leah V. Steyn, Charles W. Putnam.

**Funding acquisition:** Klearchos K. Papas.

**Investigation:** Leah V. Steyn, Delaney Drew, Demetri Vlachos, Barry Huey, Katie Cocchi, Nicholas D. Price.

**Methodology:** Leah V. Steyn, Robert Johnson.

**Project administration:** Leah V. Steyn.

**Resources:** Delaney Drew, Demetri Vlachos, Barry Huey, Katie Cocchi, Nicholas D. Price.

**Supervision:** Klearchos K. Papas.

**Validation:** Leah V. Steyn.

**Visualization:** Charles W. Putnam.

**Writing – original draft:** Charles W. Putnam.

**Writing – review & editing:** Robert Johnson, Charles W. Putnam, Klearchos K. Papas.

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
