## [Decision Letter · Decision Letter 0]

12 Apr 2023

PONE-D-22-32238Accelerated absorption of regular insulin administered via a vascularizing permeable microchamber implanted subcutaneously in diabetic Rattus norvegicusPLOS ONE

Dear Dr. Putnam,

Thank you for submitting your manuscript to PLOS ONE. After careful consideration, we feel that it has merit but does not fully meet PLOS ONE’s publication criteria as it currently stands. Therefore, we invite you to submit a revised version of the manuscript that addresses the points raised during the review process.

There are a few questions raised by the reviewer that need to be answered by the authors. Therefore, the authors are greatly recommended to address the two questions raised by one of the reviewers.

We look forward to receiving your revised manuscript.

Kind regards,

P. Balaji, Ph D

Academic Editor

PLOS ONE

Journal Requirements:

1. To comply with PLOS ONE submissions requirements, in your Methods section, please provide additional information regarding the experiments involving animals and ensure you have included details on (1) methods of sacrifice, (2) methods of anesthesia and/or analgesia, and (3) efforts to alleviate suffering.

“I have read the journal's policy and the authors of this manuscript have the following competing interests: [R.J. and KKP are co-founders of and have financial interests in Procyon Technologies LLC.]”

Please respond by return email with your amended Competing Interests Statement and we will change the online submission form on your behalf.

Additional Editor Comments:

There are a few questions raised by the reviewer that need to be answered by the authors. Therefore, the authors are greatly recommended to address the two questions raised by one of the reviewers.

Reviewers' comments:

Reviewer's Responses to Questions

**Comments to the Author**

1. Is the manuscript technically sound, and do the data support the conclusions?

Reviewer #1: Yes

Reviewer #2: Yes

2. Has the statistical analysis been performed appropriately and rigorously? 

Reviewer #1: Yes

Reviewer #2: Yes

3. Have the authors made all data underlying the findings in their manuscript fully available?

Reviewer #1: Yes

Reviewer #2: Yes

4. Is the manuscript presented in an intelligible fashion and written in standard English?

Reviewer #1: Yes

Reviewer #2: Yes

5. Review Comments to the Author

Reviewer #1: 1. What was the reason behind to select the immunodeficient rat for the study ?

2. Why particular sex(male) being preferred for the study ?

3. Kindly specify the significant contributing factor(Increasing vascular permeability or novelty of pumping design or location of implantation or if any other factors) for rapid insulin absorption in the conclusion.

Reviewer #2: The manuscript submitted on the titled, "Accelerated absorption of regular insulin administered via a vascularizing permeable microchamber implanted subcutaneously in diabetic Rattus norvegicus" is worth publishing in the PLOS ONE journal, with further minor revision of the manuscript.

1. What are the triggers to induce vascularization in the inserted microchamber?

2. Have you observed any allergic reactions after implanting the microchamber?

6. PLOS authors have the option to publish the peer review history of their article (what does this mean?). If published, this will include your full peer review and any attached files.

Reviewer #1: No

Reviewer #2: No

---

## [Author Response · Author response to Decision Letter 0]

18 Apr 2023

Reviewers’ Comments

Reviewer #1: 

1. What was the reason behind selecting the immunodeficient rat for the study?

Because the rats were to receive human insulin, we were concerned that a potential antibody response to the human protein by the subject rat might sequester or otherwise interfere with measurements of the insulin kinetic response. The nude rodent model is often used to avoid interference by the immune system in xenogeneic systems (most commonly, in human tumor models). However, the athymic nude model has also been used to avert immune antibody responses to xeno-proteins [1]. (We have added this citation to the manuscript.)

2. Why particular sex(male) being preferred for the study?

Thank you for raising this question! Even investigators working with the streptozotocin (STZ)-induced diabetes model may not be aware that estrogen interferes with the action of STZ in inducing diabetes; hence males are more sensitive to STZ and can be induced with lower doses, thereby diminishing off-target toxicities [2]. (We have also added this citation to the manuscript.)

3. Kindly specify the significant contributing factor (Increasing vascular permeability or novelty of pumping design or location of implantation or if any other factors) for rapid insulin absorption in the conclusion.

Your suggestion is an excellent one and we have revised the Conclusion to clearly state that our data indicate that robust neovascularization abetted by the outer membrane of the IVM device is important to rapid insulin absorption; the Conclusion now reads in part: “…neovascularization surrounding the IVMs was a significant contributor to the rapidity of insulin uptake.”

Reviewer # 2:

1. What are the triggers to induce vascularization in the inserted microchamber?

Membrane microarchitecture [3, 4] is the established trigger of neovascularization surrounding our microchambers. The outer (vascularizing) membrane of the mirochamber is composed of a PTFE (i.e. Teflon), having pores of ∼5μm. These pores enable ingrowth of smaller blood vessels facilitating neovascularization, which is 80 – 100X that found surrounding a 0.02 μm PTFE membrane that does not allow cells to penetrate it [3]. (Both citations are now included in the manuscript). We have revised the opening paragraph of the Discussion to emphasize these points (see Manuscript with Track Changes).

2. Have you observed any allergic reactions after implanting the microchamber?

The devices were fashioned of PTFE in large part because of its many-years’ record of successful use in humans [5]. In fact, it is so hypo-allergenic that it has been used to wrap metal or silastic devices in patients having allergic responses to the latter materials. We did not observe any signs or symptoms of systemic or local allergic reactions to the microchambers in the rats; moreover, the histology of the devices after removal showed well-developed vascularization and integration with the host tissues. 

In addition to this “Response to the Reviewers”, we have uploaded the “Revised Manuscript with Track Changes” and the unmarked version, “Manuscript.” We appreciate your careful consideration of our responses and look forward to our report appearing in PLOS ONE!

 Sincerely,

Charles W. Putnam Klearchos K. Papas

Submitting Author Corresponding Author

References cited in response to Reviewers’ queries:

1. Bartholomaeus WN, O'Donoghue H, Reed WD. Thymus-dependence of autoantibody responses to liver specific lipoprotein in the mouse. Clin Exp Immunol. 1984;55(3):541-5. PubMed PMID: 6323073; PubMed Central PMCID: PMCPMC1535937.

2. Goyal SN, Reddy NM, Patil KR, Nakhate KT, Ojha S, Patil CR, et al. Challenges and issues with streptozotocin-induced diabetes - A clinically relevant animal model to understand the diabetes pathogenesis and evaluate therapeutics. Chem Biol Interact. 2016;244:49-63. Epub 20151202. doi: 10.1016/j.cbi.2015.11.032. PubMed PMID: 26656244.

3. Brauker JH, Carr-Brendel VE, Martinson LA, Crudele J, Johnston WD, Johnson RC. Neovascularization of synthetic membranes directed by membrane microarchitecture. J Biomed Mater Res. 1995;29(12):1517-24. doi: 10.1002/jbm.820291208. PubMed PMID: 8600142.

4. Geller RL, Loudovaris T, Neuenfeldt S, Johnson RC, Brauker JH. Use of an immunoisolation device for cell transplantation and tumor immunotherapy. Ann N Y Acad Sci. 1997;831:438-51. doi: 10.1111/j.1749-6632.1997.tb52216.x. PubMed PMID: 9616733.

5. Yorita K, Takahashi J, Tano K, Ichikawa Y, Hamaguchi N, Yasui W. Twenty years' use of expanded polytetrafluoroethylene sheet for an artificial cardiac pacemaker. Clin Case Rep. 2021;9(6):e04334. Epub 20210624. doi: 10.1002/ccr3.4334. PubMed PMID: 34194807; PubMed Central PMCID: PMCPMC8223883.

---

## [Decision Letter · Decision Letter 1]

19 Jun 2023

Accelerated absorption of regular insulin administered via a vascularizing permeable microchamber implanted subcutaneously in diabetic Rattus norvegicus

PONE-D-22-32238R1

Dear Dr. Putnam,

We’re pleased to inform you that your manuscript has been judged scientifically suitable for publication and will be formally accepted for publication once it meets all outstanding technical requirements.

Kind regards,

P. Balaji, Ph D

Academic Editor

PLOS ONE

Additional Editor Comments (optional):

All the comments / suggestion have been addressed by the Authors and hence the manuscript may be accepted for publication in its Current Form

Reviewers' comments:

Reviewer's Responses to Questions

**Comments to the Author**

1. If the authors have adequately addressed your comments raised in a previous round of review and you feel that this manuscript is now acceptable for publication, you may indicate that here to bypass the “Comments to the Author” section, enter your conflict of interest statement in the “Confidential to Editor” section, and submit your "Accept" recommendation.

Reviewer #2: All comments have been addressed

2. Is the manuscript technically sound, and do the data support the conclusions?

Reviewer #2: Yes

3. Has the statistical analysis been performed appropriately and rigorously? 

Reviewer #2: Yes

4. Have the authors made all data underlying the findings in their manuscript fully available?

Reviewer #2: Yes

5. Is the manuscript presented in an intelligible fashion and written in standard English?

Reviewer #2: Yes

6. Review Comments to the Author

Reviewer #2: (No Response)

7. PLOS authors have the option to publish the peer review history of their article (what does this mean?). If published, this will include your full peer review and any attached files.

Reviewer #2: No

---

## [Editor Report · Acceptance letter]

21 Jun 2023

PONE-D-22-32238R1 

Accelerated absorption of regular insulin administered via a vascularizing permeable microchamber implanted subcutaneously in diabetic *Rattus norvegicus*

Dear Dr. Putnam:

I'm pleased to inform you that your manuscript has been deemed suitable for publication in PLOS ONE. Congratulations! Your manuscript is now with our production department. 

Kind regards, 

on behalf of

Dr. P. Balaji 

Academic Editor

PLOS ONE